# Landscape Preference: The Role of Attractiveness and Spatial Openness of the Environment

**DOI:** 10.3390/bs13080666

**Published:** 2023-08-09

**Authors:** Marek Franěk

**Affiliations:** Faculty of Informatics and Management, University of Hradec Králové, Rokitanského 62, 500 03 Hradec Králové, Czech Republic; marek.franek@uhk.cz; Tel.: +420-49-333-2374

**Keywords:** landscape preference, spatio-cognitive dimensions, emotions

## Abstract

To live a healthy lifestyle, urban residents need contact with nature and restoration in a natural environment. Environmental psychology has investigated the types and features of natural environments that could be optimal for restoration. Using a sample of undergraduates from the Czech Republic, the present study explored whether attractive and open natural environments are liked more and perceived as more restorative than unattractive and closed environments. Furthermore, this study explored which spatio-cognitive dimensions and emotional qualities of the environments are associated with the liking and perceived restoration of the environments. It was found that attractive and open environments were liked significantly more and had a significantly higher level of perceived restoration than attractive closed environments, but in the nonattractive environments, the openness of the environments had no significant effects on liking and perceived restoration. Although we only found a significant contribution of the spatio-cognitive dimension of mystery to liking and perceived restoration, emotional qualities of the environment were a good predictor for the liking and perceived restoration of natural environments. The effects of the aesthetic qualities of images and the photographic techniques used should also be considered. The results are discussed in connection with the fact that preference for attractive landscapes may lead to an underestimation of the value of ordinary nature in neighborhoods.

## 1. Introduction

In recent decades, a gradual movement of residents from rural areas to urban centers has occurred [1]. Among the solutions to many problems linked with extensive urbanization, a healthy lifestyle requires contact with nature and restoration in a natural environment. One of the research goals of environmental psychology is to investigate the features of natural environments that are optimal for restoration. For example, an important question involves the differences in the perception of the attractiveness and restorative potential of ordinary nature from our neighborhoods vs. natural environments from attractive tourist destinations. In this context, de Groot and van den Born [2] investigated preferences for different types of landscapes in a sample of Dutch residents. The majority of respondents preferred attractive environments where the greatness and power of nature can be experienced. The preference was related to education and age; people with higher education and younger people preferred this type of landscape far more than people with less education and older people. The problem is that these preference patterns may lead to an underestimation of the restorative potential of the easily accessible domestic landscapes in a neighborhood and may lead to the need for vacations in attractive distant destinations. However, intensive and heavy tourism has negative consequences for the environment. The present study aims to explore preferences for different types of landscapes and their estimated restorative potential among a sample of undergraduates from the Czech Republic and to analyze the factors associated with these preferences.

### 1.1. Environmental Preference

One of the key distinctions between people’s environmental perceptions and preferences is the difference between natural and built scenes [3]. There is mounting evidence that Western people prefer unthreatening natural environments over urban environments [4,5,6,7]. It has been found in a number of studies that landscapes that are perceived as natural are considered more scenic than clearly human-influenced (cultural) landscapes [8,9,10,11,12,13]. Apart from evolutionary assumptions [14], an explanation for this difference in environmental preference is that natural environments support more positive (aesthetic, recreational) experiences and, thus, psychological wellbeing more generally than built ones (e.g., [3]). Research suggests that environmental preference and restoration are closely related [15,16]; moreover, higher preferences for environments seem to be associated with greater affective restoration (e.g., [16,17]).

A number of studies have explored environmental features that influence aesthetic landscape values. Existing literature suggests that plants or water can enhance aesthetic preference or mental restoration [18,19]. Flowers can improve aesthetic preference [20,21] and contribute to psychological well-being [22]. Water is an important factor that determines landscape preference [5,23,24,25,26]. Research shows differences in the degree of preference for types of water. The top-rated types are oceans [27,28], water with high mountains [29], and waterfalls [30]. Landscapes with a more structured appearance are preferred over those that are perceived as less structured [31]. Mean levels of diversity and complexity are generally preferred over highly complex or homogeneous patterns [32,33,34].

### 1.2. “Mundane” vs. Attractive Natural Scenes

De Groot and van den Born [2] investigated the preferences for different types of landscapes among a representative sample of the Dutch population. The choice was among the following types of landscape: (1) a landscape modified for human needs, (2) a scenic cultural landscape (meadows, fields, forests), (3) an unmodified natural setting, allowing the visitor many interactions, and (4) a landscape where one can experience the greatness and power of nature (sea, mountains). The majority of respondents preferred the fourth type of landscape. Interestingly, this type of landscape was preferred not only by younger people but also by middle-aged people. The preference was related to education, with people with higher education preferring the fourth type of landscape far more than people with less education.

As de Groot and van den Born’s study [2] showed, there are individual differences in preference between managed rural and wild landscapes. It seems that a place of residence is the most important factor. Rural residents often express negative attitudes toward protecting nearby wilderness areas (e.g., [35,36]) and express lower preferences for wilderness landscapes than urban residents [2,37,38]. Preferences for natural landscapes also vary with age [37,39,40]. Older people display low preferences for wild nature that may be explained by their greater physical vulnerability, which may make them more sensitive to the various dangers of wilderness areas. Moreover, socioeconomic status is another factor that influences preferences for rural versus wild landscapes. People with high incomes and education prefer wild nature more [41,42].

### 1.3. Spatio-Cognitive Dimensions Associated with Environmental Preference

Several decades ago, in their pilot work, Kaplan and Kaplan [3] explored the dichotomy of natural vs. built environments and proposed that natural environments dispose with specific visual cognitive dimensions and therefore are liked and preferred over built ones. They proposed four basic spatio-cognitive dimensions: (1) *coherence*, or the degree to which elements of the environments are related and logically organized: the more coherent an environment is, the greater the preference for it; (2) *legibility*, or the extent to which elements allow an observer to understand the environment and its content: the more legible it is, the greater the preference; (3) *complexity*, which involves the number or diversity of elements in the environment: the more complexity there is, the greater the preference; and (4) *mystery*, or the degree to which the environment contains hidden information. These dimensions have been employed widely in environmental preference research (for a review, see [43]). Kaplan and Kaplan’s work was continued by Herzog [30,44], and in addition to these four basic dimensions, he proposed an additional spatio-cognitive dimension.

### 1.4. Openness of the Environment

Another important physical feature of the environment that influences people’s perceptions and preferences is spatial openness. Appleton’s [14] evolutionary prospect–refuge theory proposed that a landscape allowing a prospect is important for survival and is preferred. Furthermore, according to Fisher and Nasar [45], a level of prospect and refuge for a potential offender are taken into consideration when people judge how safe they feel in an environment. This was supported by a number of studies that showed that the feeling of danger from a possible attack may increase in a closed environment (e.g., [46,47,48,49]). For instance, it was shown that in tree alleys, there was greater perceived danger than in open landscapes with fields [48]. For alleys, danger was a more common reaction than perceived mystery of the environment. Simultaneously, a number of studies have confirmed that the spatial openness of the landscape is a factor that positively influences its preference. For instance, how well residents can see the landscape with an open view is a significant predictor of perceived landscape quality [50]; similarly, choices of preferred landscape may be driven by the preference for landscape openness [51,52]. Moreover, a recent investigation from the field of neuroscience showed that spatial information about “openness” plays an important role in the scene-responsive parahippocampal area in the brain, even in different scene categories (e.g., [53,54]).

### 1.5. Affective Qualities of the Environment

Physical environments possess affective qualities that induce expressions of pleasure, happiness, attraction, enjoyment, preference, etc. [55]. Landscape preference can also be predicted by diverse emotional responses [56]. The related stress reduction theory [10] emphasizes the role of the natural environment in reducing stress and the association of perception of this type of environment with positive emotional responses. According to this theory, exposure to environments with vegetation, water, and other natural features produces a response characterized by decreased physiological arousal, increases in positive affect, and decreases in negative affect. This theory has been supported by a large body of research [57,58,59,60,61]. The research was mostly directed by the circumplex model of Russell and Pratt [62], which proposed two dimensions, pleasure (joy, happiness, etc.) and arousal (interest, activity, surprise, etc.), to explain the differences in emotional responses to the physical environment.

### 1.6. The Present Study

The present study aims to contribute to research on preferred and restorative environments. The first goal was to explore the effect of the attractiveness and spatial openness of environments on people’s liking and perceived restoration. We predicted that attractive environments are liked more and perceived as more restorative than unattractive environments and that spatially open environments are liked more and perceived as more restorative than spatially closed environments. The second goal was to explore the association of visual spatio-cognitive dimensions of the environment with the likeability and perceived restoration of the environments. We predict that a high perceived level of the dimension coherence, complexity, and mystery of images will be associated with higher likeability and a higher level of perceived restoration. The third goal was to explore which emotional qualities are associated with the liking and perceived restoration of the environments. We predict that emotional categories with positive valence will be positively associated with liking and perceived restoration, while emotional categories with negative valence will be negatively associated with liking and perceived restoration. Finally, we explored gender differences in liking and perceived restoration of the selected environments without any specific prediction. This study follows de Groot’s and van den Born’s [2] investigation. While this study used a questionnaire, where particular types of landscapes were described, our investigation is based on the perception and evaluation of visual stimuli in laboratory conditions.

## 2. Study 1: Evaluation of Attractiveness of Selected Images

The perception of environmental attractiveness is somewhat individual. Thus, before exploring the effect of the attractiveness of the environments on individual’s liking and feeling of restoration, the estimated level of attractiveness of selected images was assessed. The second variable, the spatial openness/closedness of a natural scene, was defined as an open natural landscape vs. a closed scene inside a forest. The perceived level of spatial openness was not tested because we selected well-marked and distinct examples of spatial openness; therefore, openness of the environment was an unambiguous feature of the images used. 

### 2.1. Material and Methods

#### 2.1.1. Sample

Fifty-three undergraduates participated in the study. The sample comprised undergraduates between the ages of 18 and 25 (mean = 21.2, *SD* = 1.69, 27 males, 26 females). The participants were enrolled in various psychology courses, and they were students in informatics, financial management, or tourism at the University of Hradec Králové.

#### 2.1.2. Stimulus Material

The stimuli were photographs representing diverse natural settings. The intention was to present attractive or unattractive mundane natural scenes that were either open or closed. Based on these criteria, we selected 16 images from the set of visual stimuli used in our previous study [63]. They consisted of four images of attractive and open environments, four images of attractive and closed environments, four images of unattractive and open environments, and four images of unattractive and closed environments (Figure 1).

The attractive images were downloaded from the Pixabay internet server, which shares copyright-free images (https://pixabay.com/, accessed on 10 June 2023), and the unattractive images were taken by one of the authors. The attractive open images were represented by high mountains, lakes, or coastal scenery. These scenes were substantially different from the landscape of the Czech Republic, where participants of the study were living. In contrast, the unattractive open scenes represented the typical landscape of the Czech Republic. Attractive closed scenes were also downloaded from Pixabay and depicted outdoor environments of the Czech Republic, while unattractive closed scenes were taken in Czech forests. The attractive scenes were professional photographs taken and adjusted with the aim of great visual effect, while the images of unattractive scenes were taken under normal lighting conditions, without any further digital adjustment.

#### 2.1.3. Procedure

Images were presented online via Google Forms in a random order. The participants evaluated the attractiveness of the images through agreement/disagreement with the phrase “I find this environment very attractive”. Their responses were assessed on a five-point scale (1 = not at all, 5 = completely).

### 2.2. Results and Discussion

The mean ratings of individual categories of photographs are shown in Table 1. Next, a two-way ANOVA was conducted to assess the effects of the attractiveness and openness of presented images, as well as gender, on their estimated attractiveness. Attractiveness (attractive vs. unattractive) and openness (open vs. closed) were selected as within-subject factors, gender as a between-subject factor, and the estimated attractiveness as a dependent variable. There was a statistically significant two-way interaction between attractiveness and openness (*F*_1,204_ = 12.852, *p* < 0.001, partial η2 = 0.059). The main effects of the attractiveness of images had a significant effect on the estimated attractiveness (*F*_1,204_ = 62.113, *p* < 0.001, partial η2 = 0.233), but the main effects of the openness of images (*F*_1,204_ = 2.509, *p* = 0.115, partial η2 = 0.012) and gender (*F*_1,204_ = 0.059, *p* = 0.807, partial η2 = 0.001) had no significant effects on estimated attractiveness. In the attractive environments, there was a statistically significant difference in the estimated attractiveness between open (mean = 4.44, *SD* = 0.48) and non-open (mean = 3.85, *SD* = 0.84) environments, with a mean difference of 0.559 (95% CI, 0.162 to 1.030) estimated attractiveness points. In the nonattractive environments, estimated attractiveness was not statistically significantly different in the open environments (mean = 3.12, *SD* = 0.88) compared to non-open environments (mean = 3.35, *SD* = 1.04), with a difference of −0.23 (95% CI, −0.663 to 0.200) estimated attractiveness points.

The results revealed that the images predicted to be attractive were rated higher in attractiveness. The attractive images were estimated to be significantly more attractive than the unattractive images when they represented open environments, but the difference in the attractiveness ratings of attractive and unattractive environments was not significant if they represented closed environments.

## 3. Study 2: The Effect of the Attractiveness and Openness of Images on Their Likeability and Perceived Restoration

The study investigated the effect of the attractiveness and spatial openness of environments on their likeability and perceived restoration. Furthermore, associations of visual spatio-cognitive dimensions and emotional qualities of the environment with the liking and perceived restoration of the environments were explored.

### 3.1. Material and Methods

#### 3.1.1. Sample

Fifty-one undergraduates participated in the experiment. The sample comprised undergraduates between the ages of 19 and 25 (mean = 20.9, *SD* = 1.28, 29 males, 22 females). The participants were enrolled in various psychology courses, and they were students in informatics, financial management, or tourism at the University of Hradec Králové. The sample was not identical to the sample from Study 1.

#### 3.1.2. Procedure

The study was conducted in a laboratory. After arrival, the participants signed the informed consent form. Afterward, the following instruction appeared on the computer screen: “You will see various natural images. Look at them, try to imagine that you are in this natural environment right now, and describe your perceptions and feelings by the statements, which will be given under each image”. Sixteen images tested in Study 1 were presented. The stimuli were presented in a random order on a computer screen with a 1366 × 768-pixel resolution screen and a diagonal of 38 cm.

#### 3.1.3. Measures

A questionnaire registered assessment of (1) three spatio-cognitive dimensions: coherence, complexity, and mystery; (2) four basic emotions: joy, sadness, surprise, and fear; and finally, (3) whether the participants liked the image and whether (4) the image represented a restorative environment. Each item was assessed on a five-point scale (1 = not at all, 5 = completely). The questionnaire was presented to the participants on a computer screen. The data are available in the Appendix A: dataset.

Spatio-cognitive dimensions: The item “The individual features of this place are in harmony; they belong together” was related to the spatio-cognitive dimension of *coherence*, the item “This place contains a large number of various elements” was related to the spatio-cognitive dimension of *complexity*, and the item “I would like to explore this interesting place more” was related to the spatio-cognitive dimension of *mystery*. These items were selected from the study by Herzog and Bosley [30].Emotions: Two items were related to the pleasure dimension. The item “I feel happy here” was related to *joy* and the item “This is a pretty sad place” was related to *sadness*. The next two items were related to the arousal dimension. The item “I feel amazed here” was related to *surprise* and the item “This place scares me a little” was related to *fear*.Liking: Liking was assessed by the item “I like this place”.Restoration: Restoration was assessed by the item “This environment offers relaxation, calming and an escape from everyday stress.”

### 3.2. Results

#### 3.2.1. Influence of Attractiveness and Openness on Liking the Images

Two-way ANOVA was conducted to assess the effects of attractiveness, openness, and gender on liking the images. Attractiveness (attractive vs. unattractive) and openness (open vs. closed) were selected as within-subject factors, gender was selected as a between-subject factor, and liking was chosen as a dependent variable. There was a statistically significant two-way interaction between attractiveness and openness (*F*_1,200_ = 8.426, *p* < 0.05, partial η2 = 0.400) and an almost statistically significant two-way interaction between attractiveness and gender (*F*_1,200_ = 7.729, *p* = 0.059, partial η2 = 0.030). The main effects of attractiveness (*F*_1,200_ = 73.449, *p* < 0.001, partial η2 = 0.277) and openness (*F*_1,200_ = 8.117, *p* < 0.01, partial η2 = 0.039) were significant, but the main effect of gender (*F*_1,200_ = 0.902, *p* = 0.343, partial η2 = 0.004) was not significant. In the attractive environments, there was a statistically significant difference in liking between open (mean = 4.130, *SD* = 0.69) and non-open (mean = 3.572, *SD* = 0.67) environments, with a mean difference of 0.558 (95% CI, 0.189 to 0.927) liking evaluation points. In the nonattractive environments, liking was not statistically significantly different in the open environments (mean = 2.995, *SD* = 0.74) compared to non-open environments (mean = 3.000, *SD* = 0.73), with a difference of −0.005 (95% CI, −0.037 to 0.364) liking evaluation points. Thus, attractiveness and openness had a combined effect, and attractive and open environments were liked more than attractive closed environments. However, in nonattractive environments, the openness/closedness of the environment had no effect on liking. Moreover, an almost significant interaction between gender and attractiveness showed that, while in the liking of attractive images there was no difference between males and females, females (mean = 3.192, *SD* = 0.69) liked unattractive images more than males (mean = 2.830, *SD* = 0.73), with a mean difference of 0.362 (95% CI, −0.749 to 0.025) liking evaluation points.

#### 3.2.2. Influence of Attractiveness and Openness on Perceived Restoration

Two-way ANOVA was conducted to assess the effects of attractiveness and openness on the perceived restoration of the environment that the images contained. Attractiveness (attractive vs. unattractive) and openness (open vs. closed) were within-subject factors, gender was a between-subject factor, and perceived restoration was chosen as a dependent variable. There was a statistically significant two-way interaction between attractiveness and openness (*F*_1,200_ = 12.436, *p* < 0.001, partial η2 = 0.059). The main effects of attractiveness (*F*_1,200_ = 66.341, *p* < 0.001, partial η2 = 0.249) and openness (*F*_1,200_ = 9.260, *p* < 0.01, partial η2 = 0.044) were also significant. The main effect of gender (*F*_1,200_ = 0.320, *p* = 0.572, partial η2 = 0.002) was not significant. In the attractive environments, there was a statistically significant difference in restoration between open (mean = 4.111, *SD* = 0.51) and non-open (mean = 3.538, *SD* = 0.55) environments, with a mean difference of 0.572 (95% CI, 0.249 to 0.895) evaluation points. In the nonattractive environments, restoration was not statistically significantly different in the open environments (mean = 3.091, *SD* = 0.72) compared to non-open environments (mean = 3.013, *SD* = 0.66), with a mean difference of −0.043 (95% CI, −0.366 to 0.279) evaluation points. Thus, attractiveness and openness had a combined effect, and attractive and open environments had a higher level of perceived restoration than attractive closed environments. However, in nonattractive environments, the openness/closedness of the environment had no effect on perceived restoration.

#### 3.2.3. Associations between Spatio-Cognitive Dimensions and Environment Liking

Multiple regressions were carried out to find the contributions of spatio-cognitive dimensions of coherence, complexity, mystery, and gender to image liking. The calculations were conducted separately for four types of environments—attractive open, attractive closed, unattractive open, and unattractive closed. The results (Table 2) showed that only the dimension of mystery significantly contributed to liking, but only in attractive closed (*R*^2^ = 0.196, *F*_4,47_ = 2.863, *p* < 0.05), unattractive open (*R*^2^ = 0.343, *F*_4,47_ = 6.141, *p* < 0.001), and unattractive closed environments (*R*^2^ = 0.386, *F*_4,7_ = 7.395, *p* < 0.05).

#### 3.2.4. Associations between Spatio-Cognitive Dimensions and Perceived Restoration

Multiple regressions were carried out to find the contributions of the spatio-cognitive dimensions of coherence, complexity, mystery, and gender to perceived restoration. The calculations were conducted separately for different types of environments—attractive open, attractive closed, unattractive open, and unattractive closed. Similar to the previous analysis, the results (Table 3) showed that only the dimension of mystery significantly contributed to perceived restoration, but only in unattractive open (*R*^2^ = 0.277, *F*_4,47_ = 4.490, *p* < 0.01) and unattractive closed environments (*R*^2^ = 0.307, *F*_4,47_ = 5.214, *p* < 0.01).

#### 3.2.5. Associations between Emotional Categories and Environment Liking

Multiple regressions were carried out to find the contributions of the emotional categories of joy, surprise, fear, sadness, and gender to image liking. The calculations were conducted separately for four types of environments—attractive open, attractive closed, unattractive open, and unattractive closed (Table 4). In attractive open environments, the results showed a statistically significant association between the emotional category of surprise and liking of images (*R*^2^ = 0.403, *F*_5,46_ = 6.337, *p* < 0.001). In attractive closed environments, the results also showed a statistically significant association between the emotional category of surprise and the liking of images (*R*^2^ = 0. 365, *F*_5,46_ = 5.288, *p* < 0.001). In the unattractive open environment, the results showed a statistically significant association between the emotional categories of joy and surprise and the liking of images (*R*^2^ = 0.640, *F*_5,46_ = 16.284, *p* < 0.001). In an unattractive closed environment, the results showed a statistically significant association between the emotional category of joy and the liking of images (*R*^2^ = 0.626, *F*_5,46_ = 15.426, *p* < 0.001).

#### 3.2.6. Associations between Emotional Categories and Perceived Restoration

Multiple regressions were carried out to find the contributions of the emotional categories of joy, surprise, fear, sadness, and gender to perceived restoration. The calculations were conducted separately for four types of environments—attractive open, attractive closed, unattractive open, and unattractive closed (Table 5). In attractive open environments, the results showed a statistically significant positive association between the emotional category of surprise and perceived restoration and a statistically significant negative association between the emotional category of fear and perceived restoration (*R*^2^ = 0.686, *F*_5,46_ = 20.516, *p* < 0.001). In attractive closed environments, the results showed a statistically significant association between the emotional category of joy and perceived restoration (*R*^2^ = 0.553, *F*_5,46_ = 11.3703, *p* < 0.001). In the unattractive open environments, the results showed a statistically significant positive association between the emotional categories of joy and sadness and perceived restoration and an almost significant (*p* = 0.054) negative association between fear and perceived restoration (*R*^2^ = 0.685, *F*_5,46_ = 20.024, *p* < 0.001). In an unattractive closed environment, the results showed a statistically significant positive association between the emotional categories of joy and sadness and perceived restoration and a significant negative association between the emotional category of fear and perceived restoration (*R*^2^ = 0.710, *F*_5,46_ = 22.473, *p* < 0.001).

## 4. Discussion

The present study aimed to explore whether attractive environments are liked more and perceived as more restorative than unattractive environments and whether spatially open environments are liked more and perceived as more restorative than spatially closed environments. Furthermore, we explored which spatio-cognitive dimensions and emotional qualities of an environment are associated with the likeability and perceived restoration of the environment.

We found that the effect of the attractiveness of the environment mediated the effect of openness. Attractive and open environments were liked significantly more and had a significantly higher level of perceived restoration than attractive closed environments and both unattractive environments, but in the nonattractive environments, the openness of the environments had no significant effects on liking and perceived restoration. In accordance with these findings, the attractive images were perceived as significantly more attractive than unattractive images, but only if they represented open environments.

However, the preference for landscape openness is likely to be a more complex issue. For instance, certain findings show that liking for open landscapes can be culturally dependent. A more recent study [64] that compared landscape preferences among Swedish and non-Western samples (East Timor, Malaysia, Colombia, Venezuela, and Guianas) showed that participants from the non-Western samples favored more forested settings, i.e., a low degree of openness. Furthermore, it seems that mental states can also influence landscape preferences in terms of their naturalness and openness. People in a fatigued state [65] have a greater preference for landscapes with a higher naturalness and those in an indignant state have a greater preference for relatively private, less-open landscapes. Our data are consistent with the findings by de Groot and van den Born [2], who showed that the majority of Dutch respondents preferred an attractive environment, where the greatness and power of nature can be experienced. Similarly, our respondents preferred similar environments over unattractive ones. Our images of attractive open environments consisted of photographs of high mountains, rocks, or seashores with high rocks, while attractive closed environments represented scenes amidst dense vegetation and forests. In contrast, both unattractive open and closed images represented scenes from common, slightly undulating landscapes of central areas of the Czech Republic. Consistently, de Groot and van den Born [2] registered that younger people with higher education preferred this type of landscape more than people with less education and older people. Our participants were university undergraduates.

Although environmental preference studies did not reveal systematic gender difference, our data in Study 2 showed that females tended to like unattractive images more than males did. However, the effect was small and was at the border of the conventionally respected value of statistical significance (*p* = 0.059). This may be explained in terms of traditional gender stereotypes—males prefer more adventures provided by attractive landscapes (e.g., climbing in high mountains) than females. Clearly, because the effect was small and almost significant, this finding can reflect individual differences in the specific sample and cannot be generalized.

It is worth considering how the environment in which people live can influence their environmental preferences. The Netherlands has a flat landscape, and therefore, it could be predicted that people may wish to spend their holidays and vacations in entirely different landscapes. While the landscape of the Czech Republic is undulating and more varied, although it lacks very high mountains, it is mostly not considered to be an attractive environment where “the greatness and power of nature can be experienced”. In addition, the participants of our study mostly live in a flat and rural area of the country where the university is located. Therefore, it would be interesting to explore landscape preferences for the restoration of people living in substantially different landscapes, for instance, in the wilderness, high mountains, or attractive coastal areas.

Furthermore, we investigated which spatio-cognitive dimensions and emotional qualities are associated with the liking and perceived restoration of the environments. We explored associations with basic spatio-cognitive dimensions of coherence, complexity, and mystery proposed by Kaplan and Kaplan [3] that should be associated with environmental preference. However, more recently, Joye and Dewitte [66] argued that the framework of this theory, although it has been highly influential in the field of restoration studies, has important empirical and conceptual shortcomings which require further empirical elaboration. It is not surprising that our study did not fully confirm that the liking and perceived restoration of the environments should be associated with these dimensions. We only found a significant contribution of the spatio-cognitive dimension of mystery for liking and perceived restoration in both unattractive open and closed environments. Thus, the mystery of “mundane”, ordinary natural environments increases their liking and perceived restoration. Although we did not test the association between attractiveness and mystery, it is possible that some form of mystery increases the attractiveness of unattractive environments, making them more attractive for restoration purposes.

In contrast, the emotional qualities of the environment were good predictors for liking and perceived restoration of natural environments. It is not surprising that the emotions of joy and surprise are associated positively with liking and perceived restoration and that the emotion of fear was negatively associated with these metrics. Curiously, sadness was associated with perceived restoration in both types of unattractive environments but not with liking the environment. Perhaps sadness was associated with some form of nostalgia, perhaps with memories of experiences from the neighboring environment from childhood. However, this association needs to be more deeply analyzed, and our data cannot provide a clearly convincing explanation.

It is worth commenting on the extent to which the photographic techniques used and the visual quality of the photographs themselves could influence an individual’s liking of presented images. The study by de Groot and van den Born [2] used a questionnaire where the landscapes were described verbally, which is an appropriate method in a sociological survey but not in a study based on visual perception. In an ideal situation, the investigation of environmental preference should be conducted in situ. Although there are studies that investigate participants’ reactions in situ, either comparing differences between urban and natural walks (e.g., [58,67]) or forest experiences (e.g., [68,69]), this research procedure is not very frequent, and for understandable reasons, it is not applicable in this type of study. For a long time, environmental preference research has used environmental simulations, from simple black-and-white photos to color slides, videos, and virtual reality. Stamps [70], in his meta-analysis carried out in the 1990s, concluded that the different simulations (drawings and black-and-white and color photographs) produced very similar preferences. Environmental simulations in various forms are still frequently used in environmental psychology research (for a review, see [71]); however, the issue of the aesthetic quality of environmental simulations and its influence on preference has not been adequately explored. There is only one study by Tinia and Leder [72], in which the authors performed manipulations of photographs of natural and built environments to verify whether image quality influenced environment preference. A degraded version of a photograph was produced by manipulations of the sharpness, noise and grain, contrast, color fidelity, and color saturation. They found that the visual quality of the images could influence preference judgments, but it had this effect for built scenes rather than natural ones. Specifically, built scenes of high image quality were liked more than natural scenes of low image quality. This shows that the quality of a photograph can influence an individual’s evaluation of an environment. On the other hand, a photograph is only a representation of a specific environment, which a person can imagine in its real form. That is why we gave the instruction “Try to imagine that you are in this natural environment right now, and describe your perceptions and feelings”. This should, at least, partially ensure that the participants did not describe their perception of the specific photograph, but the environment which is represented by the photographs. However, the influence of visual quality of the environmental simulation is a question that should be further investigated. Finally, we considered the problem that a preference for attractive nature that can only be reached by traveling to distant tourist destinations may lead to the need for vacations in attractive distant destinations. Moreover, it might lead to underestimation of the restorative potential of easily accessible domestic landscapes in a neighborhood. Clearly, “heavy” tourism has negative consequences for the environment. In this context, several decades ago, Knighton [73] called the tendency to prefer attractive nature and underestimate the value of ordinary nature in one’s own neighborhood “eco-porn”. Similarly, Levi and Kocher [74] documented that this tendency may result in support for the protection of attractive natural environments in national parks but, on the other hand, a decrease in interest in the natural environment where one resides.

The findings from recent post-industrial societies revealed that school children are not able to identify even the most common plants (e.g., [75]) and, curiously, they are sometimes more familiar with exotic flagship species than with their local fauna (e.g., [76]). This is the challenge for various forms of environmental education, although on the other hand, the effect of environmental education can be reduced by the rapid development of modern digital technologies, namely in many members of the younger generation. Thus, the implication is to further investigate environmental preferences, including the factors that influence them, and explore how to promote the value of local natural landscapes, even if they are not as attractive as others.

This study has several limitations. The research was conducted with only a small sample of young people who were university students; thus, it does not give information about the preferences of the whole population of one country, as the study by de Groot and van den Born [2] provided. On the other hand, the research (Study 2) was conducted individually in laboratory conditions, which enabled us to control possible side effects, but limited the number of participants. Although a student sample might be a limitation, the number of undergraduates in the population is gradually increasing; they are an active social group and, moreover, their preferences for future life are formed in their youth.

A further limitation could also be the small number of images used in our study. On the other hand, after an extensive number of investigations of environmental preference and the effects of natural environments, specific types of environments, their perception, evaluation, and preference have already been identified; thus, we suppose that the small number of visual stimuli in our study has not been an essential drawback.

## Figures and Tables

**Figure 1 behavsci-13-00666-f001:**
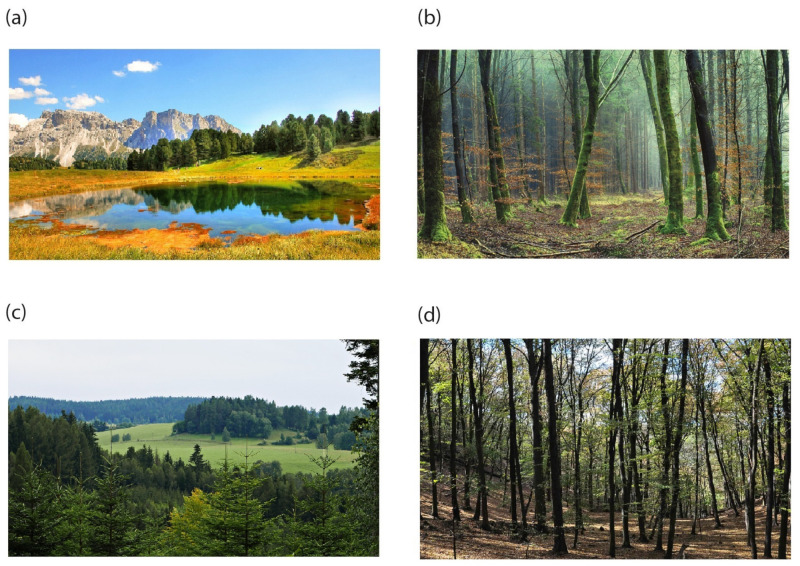
Examples of the stimulus material: (**a**) attractive open environment, (**b**) attractive closed environment, (**c**) unattractive open environment, and (**d**) unattractive closed environment.

**Table 1 behavsci-13-00666-t001:** Mean rating of individual categories of photographs. The scale ranged from 1 to 5 (1 = not at all, 5 = completely).

	Open		Closed	
	Mean	*SD*	Mean	*SD*
Attractive	4.44	0.48	3.85	0.84
Unattractive	3.12	0.88	3.35	1.04

**Table 2 behavsci-13-00666-t002:** Multiple regression results for environmental liking. Predictors: spatio-cognitive dimensions of coherence, complexity, mystery, and gender. Conducted separately for attractive open, attractive closed, unattractive open, and unattractive closed environments. Statistically significant results are indicated in bold.

	B	SE	β	*t*	*p*
**Attractive open**					
Coherence	−0.046	0.143	−0.068	−0.3020	0.750
Complexity	0.033	0.152	0.037	0.215	0.831
Mystery	0.079	0.154	0.184	0.510	0.612
Gender	−0.071	0.145	−0.100	−0.480	0.626
*R* ^2^	0.018				
Adj *R*^2^	−0.064				
*SE*	0.711				
*F* _4,47_	0.218 *p* = 0.927				
**Attractive closed**					
Coherence	0.008	0.132	0.013	0.067	0.947
Complexity	0.207	0.163	0.172	1.336	0.212
Mystery	**0.275**	**0.165**	**0.221**	**1.670**	**0.101**
Gender	−0.037	0.140	−0.050	−0.265	0.792
*R* ^2^	0.196				
Adj *R*^2^	0.128				
*SE*	0.626				
*F* _4,47_	2.863 *p* < 0.05				
**Unattractive open**					
Coherence	−0.023	0.130	−0.030	−0.181	0.857
Complexity	0.264	0.183	0.228	1.444	0.155
Mystery	**0.344**	**0.177**	**0.349**	**0.9471**	**0.005**
Gender	0.154	0.124	0.228	1.243	0.220
*R* ^2^	0.343				
Adj *R*^2^	0.287				
*SE*	0.626				
*F* _4,47_	6.141 *p* < 0.001				
**Unattractive closed**					
Coherence	−0.080	0.122	−0.010	−0.654	0.516
Complexity	0.182	0.170	0.152	1.070	0.290
Mystery	**0.456**	**0.166**	**0.469**	**2.750**	**0.008**
Gender	0.169	0.118	0.245	1.432	0.159
*R* ^2^	0.386				
Adj *R*^2^	0.334				
*SE*	0592				
*F* _4,47_	7.395 *p* < 0.05				

**Table 3 behavsci-13-00666-t003:** Multiple regression results for perceived restoration. Predictors: spatio-cognitive dimensions of coherence, complexity, mystery, and gender. Conducted separately for attractive open, attractive closed, unattractive open, and unattractive closed environments. Statistically significant results are indicated in bold.

	B	SE	β	*t*	*p*
**Attractive open**					
Coherence	0.076	0.137	0.084	0.556	0.581
Complexity	0.085	0.145	0.072	0.591	0.557
Mystery	0.224	0.146	0.199	1.530	0.132
Gender	−0.137	0.139	0.140	−0.990	0.329
*R* ^2^	0.107				
Adj *R*^2^	0.032				
*SE*	0.502				
*F* _4,47_	1.430 *p* = 0.238				
**Attractive closed**					
Coherence	0.206	0.138	0.250	1.666	0.141
Complexity	0.153	0.170	0.117	0.897	0.374
Mystery	0.185	0.172	0.123	1.079	0.286
Gender	0.011	0.144	0.012	0.080	0.936
*R* ^2^	0.128				
Adj *R*^2^	0.073				
*SE*	0.535				
*F* _4,47_	1.721 *p* = 0.161				
**Unattractive open**					
Coherence	0.056	0.137	0.067	0.407	0.686
Complexity	0.098	0.192	0.082	0.511	0.610
Mystery	**0.442**	**0.186**	**0.432**	**2.381**	**0.021**
Gender	0.004	0.130	0.006	0.032	0.975
*R* ^2^	0.277				
Adj *R*^2^	0.215				
*SE*	0.634				
*F* _4,47_	4.490 *p* < 0.01				
**Unattractive closed**					
Coherence	0.014	0.130	0.015	0.108	0.915
Complexity	0.084	0.181	0.064	0.464	0.644
Mystery	**0.490**	**0.176**	**0.460**	**2.784**	**0.008**
Gender	0.003	0.125	0.004	0.027	0.979
*R* ^2^	0.307				
Adj *R*^2^	0.248				
*SE*	0.575				
*F* _4,47_	5.214 *p* < 0.01				

**Table 4 behavsci-13-00666-t004:** Multiple regression results for environmental liking. Predictors: emotional categories of joy, surprise, fear, sadness, and gender. Conducted separately for attractive open, attractive closed, unattractive open, and unattractive closed environments. Statistically significant results are indicated in bold.

	B	SE	β	*t*	*p*
**Attractive open**					
Joy	0.308	0.169	0.360	1.182	0.075
Surprise	**0.410**	**0.157**	**0.460**	**2.613**	**0.012**
Fear	0.0476	0.163	0.051	0.84	0.777
Sadness	0.033	0.170	0.038	0.194	0.846
Gender	−0.022	0.127	−0.030	−0.170	0.866
*R* ^2^	0.403				
Adj *R*^2^	0.339				
*SE*	0.560				
*F* _5,46_	6.337 *p* < 0.001				
**Attractive closed**					
Joy	0.321	0.173	0.350	1.889	0.069
Surprise	**0.375**	**0.146**	**0.340**	**2.565**	**0.017**
Fear	−0.004	0.224	−0.032	−0.191	0.849
Sadness	0.041	0.206	0.031	0.199	0.842
Gender	−0.052	0.135	−0.071	−0.359	0.699
*R* ^2^	0.365				
Adj *R*^2^	0.296				
*SE*	0.562				
*F* _5,46_	5.288 *p* < 0.001				
**Unattractive open**					
Joy	**0.573**	**0.127**	**0.579**	**4.500**	**0.000**
Surprise	**0.277**	**0.129**	**0.277**	**2.149**	**0.037**
Fear	−0.109	0.120	−0.169	−0.913	0.366
Sadness	0.164	0.120	0.2310	1.365	0.178
Gender	0.124	0.090	0.184	1.374	0.176
*R* ^2^	0.640				
Adj *R*^2^	0.600				
*SE*	0.469				
*F* _5,46_	16.284 *p* < 0.001				
**Unattractive closed**					
Joy	**0.560**	**0.130**	**0.632**	**4.593**	**0.000**
Surprise	0.234	0.131	0.227	1.784	0.081
Fear	−0.115	0.131	−0.184	−0.873	0.387
Sadness	0.172	0.130	0.200	1.330	0.1920
Gender	0.174	0.094	0.253	1.851	0.071
*R* ^2^	0.626				
Adj *R*^2^	0.586				
*SE*	0.467				
*F* _5,46_	15.426 *p* < 0.001				

**Table 5 behavsci-13-00666-t005:** Multiple regression results for perceived restoration. Predictors: emotional categories of joy, surprise, fear, sadness, and gender. Conducted separately for attractive open, attractive closed, unattractive open, and unattractive closed environments. Statistically significant results are indicated in bold.

	B	SE	β	*t*	*p*
**Attractive open**					
Joy	**0.700**	**0.122**	**0.605**	**5.684**	**0.000**
Surprise	−0.078	0.114	−0.065	−0.688	0.494
Fear	**−0.424**	**0.118**	**−0.348**	**−3.600**	**0.001**
Sadness	0.200	0.123	0.170	1.0624	0.111
Gender	−0.171	0.092	−0.175	−1.846	0.071
*R* ^2^	0.686				
Adj *R*^2^	0.652				
*SE*	0.301				
*F* _5,46_	20.516 *p* < 0.001				
**Attractive closed**					
Joy	**0.663**	**0.145**	**0.600**	**4.576**	**0.000**
Surprise	0.194	0.123	0.107	1.581	0.121
Fear	0.276	0.188	0.169	1.469	0.149
Sadness	−0.149	0.173	−0.093	−0.860	0.394
Gender	−0.006	0.113	−0.007	−0.055	0.956
*R* ^2^	0.553				
Adj *R*^2^	0.504				
*SE*	0.386				
*F* _5,46_	11.370 *p* < 0.001				
**Unattractive open**					
Joy	**0.773**	**0.119**	**0.754**	**6.500**	**0.000**
Surprise	0.073	0.120	0.070	0.604	0.549
Fear	−0.221	0.112	−0.310	−1.977	0.054
Sadness	**0.310**	**0.112**	**0.374**	**2.762**	**0.008**
Gender	−0.019	0.084	−0.028	−0.230	0.819
*R* ^2^	0.685				
Adj *R*^2^	0.651				
*SE*	0.422				
*F* _5,46_	20.024 *p* < 0.001				
**Unattractive closed**					
Joy	**0.728**	**0.115**	**0.704**	**6.4352**	**0.000**
Surprise	0.147	0.116	0.130	1.274	0.210
Fear	**−0.237**	**0.116**	**−0.346**	**−2.040**	**0.047**
Sadness	**0.358**	**0.114**	**0.377**	**3.133**	**0.003**
Gender	0.025	0.083	0.033	0.300	0.768
*R* ^2^	0.710				
Adj *R*^2^	0.78				
*SE*	0.376				
*F* _5,46_	22.473 *p* < 0.001				

## Data Availability

The datasets supporting this article have been uploaded as part of the Appendix A.

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
