# Peer review of "Landscape Preference: The Role of Attractiveness and Spatial Openness of the Environment"

_behavsci, 2023, doi:10.3390/bs13080666_

Round 1

Reviewer 1 Report

This paper explores the influences of attractiveness, openness, spatio-cognitive dimensions and emotional categories on environment liking and perceived restoration. My comments are as follows.

1. As stated in the Introduction section, there are literatures that confirms the influences of attractiveness, openness, spatio-cognitive dimensions and emotional categories on perceived restoration. This paper provides a study using undergraduates as experimental subjects, while undergraduates are single in terms of population characteristics, so it is difficult for the paper to report more abundant findings. The paper needs to propose more in-depth research and innovations.

2. In the second section, what is the basis for dividing the types of 16 images? Do the authors classify the images, or is it classified based on the subject's subjective perception? What is the method of division?

3. There are experiments both in the second the third section. Are the undergraduates in the two experiments the same? Why are the numbers of subjects different?

4. The paper uses ANOVA and multiple regression models to analyze the influences of attractiveness, openness, spatio-cognitive dimensions and emotional categories on environment liking and perceived restoration. The two methods can not control the influence of other variables, such as population characteristics. This may explain why sadness has a negative effect on perceived restoration. It is suggested to add more control variables.

Author Response

Dear reviewer, thank you very much for a time you spent with reading of our manuscript and your comments.

  1. This paper provides a study using undergraduates as experimental subjects, while undergraduates are single in terms of population characteristics, so it is difficult for the paper to report more abundant findings. The paper needs to propose more in-depth research and innovations.

We agree that the contribution of the present study to current knowledge is not substantial. But, in our opinion, the topic was not adequately explored. To date, there are number of studies, which come from different countries and substantially different landscapes (e.g. Netherland, Sweden, China) were analyzed. Moreover, sometimes the results are mixed.

Very extensive study by Groot and van den Born’s study worked with a representative sample of the Dutch population, but it used a questionnaire, where types of landscapes were described verbally (we added this information, lines 153-155). The approach of environmental sociology differs from laboratory experimental research with perception and evaluation of visual stimuli, where number of participants and composition of the sample should be from many reasons limited. We discussed this limitation in the discussion section, this part of the original submission was enlarged (lines 500-505).

In the study we tried to conduct more in-depth analyses, to explore join effect of attractiveness and openness and association between of spatio-cognitive and emotional dimensions to liking and perceived restorative potential. In contrast to Attention restoration theory, we found that spatio-cognitive dimensions coherence and complexity did not influence liking and feeling of restoration. It can contribute to recent discussions about validity this theory.

  1. In the second section, what is the basis for dividing the types of 16 images? Do the authors classify the images, or is it classified based on the subject's subjective perception? What is the method of division?

The images were selected from the set of stimuli used in our previous study FranÄ›k, M.; Petružálek, J.; Šefara, D. Facial Expressions and Self-Reported Emotions When Viewing Nature Images. Int. J. Environ. Res. Public Health 2022, 19,10588.

They were selected and classified by the author of the study. There are two characteristics: openness and attractiveness. Spatial openness/closedness of natural scene was unequivocally defined as an open landscape vs. a closed scene inside a forest.  However, attractiveness of natural scene may be somewhat subjective, therefore, prior to the main study, we tested this variable in Study 1. Because the selected images differed in perceived attractiveness according to our prediction, they were used in Study 2 as representatives of attractive/unattractive natural scenes.

We added this information to the sections 2.0 and 2.1.2

  1. There are experiments both in the second the third section. Are the undergraduates in the two experiments the same? Why are the numbers of subjects different?

The sample in the Study 1 was not identical with the sample in Study 1. It was already mentioned in the original submission, lines 233-234. Considerations about the attractiveness of the photographs could influence estimation of their liking and perceived restoration, therefore in Study 2 we had a different sample.

  1. The paper uses ANOVA and multiple regression models to analyze the influences of attractiveness, openness, spatio-cognitive dimensions and emotional categories on environment liking and perceived restoration. The two methods cannot control the influence of other variables, such as population characteristics. This may explain why sadness has a negative effect on perceived restoration. It is suggested to add more control variables.

We added to ANOVA and multiple regression models also variable gender and found some effect of gender. However, the student sample is very consisted in terms of age, economic level, social status, and place of residence, thus the effect of these variables was not explored.

Reviewer 2 Report

The work is interesting, but the chosen sample of a very small group of students has limited impact. Sometimes the paper say obvious sentences about attractiveness, which shold be avoided as: The results revealed that the attractive images were rated higher in attractivity. It become more interesting when the focus is shifted to its relation with openness.

English Language is adequate for the purpose of the work.

Author Response

Dear reviewer, thank you very much for a time you spent with reading of our manuscript and your comments.

The work is interesting, but the chosen sample of a very small group of students has limited impact.

We added to the discussion (lines 500-505): „On the other hand, the research (Study 2) was conducted individually in laboratory conditions, which enables to control possible side effects, but, limits number of participants.“

Sometimes the paper say obvious sentences about attractiveness, which shold be avoided as: The results revealed that the attractive images were rated higher in attractivity.

The sentence was corrected.

 It become more interesting when the focus is shifted to its relation with openness.

We added discussion about the effect of opennes, lines 387- 395.

Reviewer 3 Report

The paper suggests that there was an influence of the fact that the surveyed people treated unattractive types of landscape as common for them. It is trivial that we like to explore new places, and what we already know, even considering it as nice, we consider less attractive. In The Netherlands, everything that has been kept natural is restricted to legally protected areas. The artificiality of the landscape, where the avenues of trees are composed of identical specimens of the same size and shape, where the emerald meadows are not even smeared with random buttercups and daisies, must have to do with the appreciation of otherness represented by free succession and chance.

From the work of de Groot and van den Born, it can be inferred that higher education similarly promotes environmental awareness and appreciation of the natural landscape. Higher environmental awareness filters out artificial treatments of photographs, turning to biodiversity as a highly influential feature.

The work is an interesting contribution to research on human preferences for the landscape. However, the results are reliable only to a certain extent, because they are strongly influenced by the photographic technique, which significantly blurs the reactions that would arise in the presence of a direct presence in the field. Differences in the technique of image processing are, however, underlined in 2.1.2. Stimulus material. In my opinion, the effect of this technique on the results is greater than was presented in the paper. This requires a clear indication that the authors have omitted this aspect and an explanation as to why. The work does not refer to the impact of photo processing and their artistic features, which must have influenced the reaction of the respondents. Here, the processing of the photo, image frame, weather conditions and color always have a huge impact. The pictures, called "Attractive examples" (Figure 1), had geometric elements of structure emphasized, as in Pop Art paintings. The characteristics of the attractiveness of the photos and their impact on the responses of the respondents were not specified.

Appreciation of the natural landscape is always associated with education in biology. Such people give a higher mark to natural plant communities, which they know from the degree of diversity, smell, seasonal variability. Other people refer to simple aesthetics unrelated to naturalness.

The authors are aware that their work does not give unambiguous results and explain this in the Discussion

I suggest adding a few sentences about the aesthetics of the photos and the lack of impact of direct contact with the places visible in the photos on the respondents. These are acceptable simplifications, but they require a broader commentary, because they strongly affect the results.

Author Response

Dear reviewer, thank you very much for a time you spent with reading of our manuscript and your comments. We carefully considered your comment regarding the influence of aesthetics of the photos. We added relevant literature to the topic and discussed this limitation.

Reviewer: In my opinion, the effect of this technique on the results is greater than was presented in the paper. This requires a clear indication that the authors have omitted this aspect and an explanation as to why. The work does not refer to the impact of photo processing and their artistic features, which must have influenced the reaction of the respondents. Here, the processing of the photo, image frame, weather conditions and color always have a huge impact. 

I suggest adding a few sentences about the aesthetics of the photos and the lack of impact of direct contact with the places visible in the photos on the respondents. These are acceptable simplifications, but they require a broader commentary, because they strongly affect the results.

Answer: We added an extensive discussion about the effect of the photographic technique and the visual quality of the photographs that could limit our results, lines 450-480.

Reviewer: Appreciation of the natural landscape is always associated with education in biology. Such people give a higher mark to natural plant communities, which they know from the degree of diversity, smell, seasonal variability. Other people refer to simple aesthetics unrelated to naturalness

Answer: We added the points that can be relevant to this problem in discussion, lines 491 – 497.